

# Channel evolution processes in a diamictic glacier foreland. Implications on downstream sediment supply: case study Pasterze / Austria

Michael Paster[1], Peter Flödl[1], Anton Neureiter[2], Gernot Weyss[2], Bernhard Hynek[2], Ulrich Pulg[3], Rannveig Ø. Skoglund[4], Helmut Habersack[1], Christoph Hauer[1]

[1]CD-Laboratory for Sediment research and management, Institute of Hydraulic Engineering and River Research, Department of Water, Atmosphere and Environment, University of Natural Resources and Life Sciences Vienna, Am Brigittenauer Sporn 3, 1200 Vienna, Austria
[2]Climate-Impact-Research, GeoSphere Austria, Hohe Warte 38, 1190 Vienna, Austria
[3]UNI Research Miljø, Laboratorium for Freshwater Ecology and Inland Fisheries, Nygårdsgaten 112, 5006 Bergen, Norway
[4]University of Bergen, Fosswinckelsgt. 6, 5006, Bergen, Norway

*Correspondence to*: *Michael Paster (michael.paster@boku.ac.at)*

**Abstract.** Global warming and glacier retreat are affecting the morphodynamics of proglacial rivers. In response to changing hydrology, the altered hydraulics will significantly impact future glacifluvial erosion and proglacial channel development. This study analyses a proglacial channel evolution process at the foreland of Austria's biggest glacier Pasterze, by predicted runoff until 2050 based on a glacio-hydrological model. A high-resolution digital elevation model was created by an unmanned aerial vehicle, sediment was sampled, a one-dimensional hydrodynamic-numerical model was generated, and bedload transport formulas were used to calculate the predicted transport capacity of the proglacial river. Due to the fine sediment composition near the glacier terminus ($d_{50}$< 79 mm), the calculation results underline the process of headward erosion in the still unaffected, recently deglaciated river section. In contrast, an armor layer is already partly established by a coarser grain size distribution in the already incised river section ($d_{50}$> 179 mm). Furthermore, already reoccurring exposed non-fluvial grain sizes combined with decreasing flow competence in the long term indicate erosion-resistant pavement layer formation disconnecting the subsurface sediments for glacifluvial reworking (vertical landform decoupling). The presented study shows that subsystems exhibiting pavement layer formation by grains exceeding the predicted transport capacity supported by non-fluvial sediments are found at the investigated glacier foreland. Thus, an extension accompanied by a refinement of the glacifluvial system in the sediment cascade approach was developed as a central result.

## 1 Introduction

Since the Little Ice Age (LIA) around 1850, global warming has caused temporal and spatial changes in high mountain areas by glacier retreat (e.g., Zemp et al., 2019; Fischer et al., 2018; Huss and Hock, 2018) and permafrost decline (Harris et al., 2009). While deglaciation of European glaciers has accelerated and repeatedly reached peak values in recent years (Sommer



et al., 2020), formerly glaciated areas are continuously expanding and are characterized by high geomorphological activity
(e.g., Avian et al., 2018; Lane et al., 2017; Carrivick et al., 2013; Cavalli et al., 2013; Gruber et al., 2004).
Deglaciated areas in direct proximity to the glacier terminus are termed proglacial (Slaymaker, 2009) and are confined by LIA
moraines (Heckmann and Morche, 2019). Within this steadily increasing spatial boundary due to glacier retreat, loose and
unconsolidated sediment exceeds the 'geological norm' defined by non-glaciated catchments. Proglacial areas are, therefore,
transitional landscapes adapting to this geological norm within the paraglacial period (Ballantyne, 2002; Church and Ryder,
1972). This adjustment occurs through various geomorphological processes (e.g., gully erosion, avalanches, debris flows),
where sediment is reworked along the gravitational gradient (Ballantyne, 2002). In contrast, continuous sediment supply is
given by (sub)glacial erosion (e.g., Alley et al., 2019; Hallet et al., 1996), where moderately well-rounded (Benn and Evans,
2013) poorly sorted unconsolidated material ranging in size from sand to cobbles up to boulders (diamictic till; Harland et al.,
1966) is deposited. The entire sediment production and reworking process chain of (temporary) sediment storages within a
catchment can be described in a sediment cascade (Chorley and Kennedy, 1971). The sediment connectivity between these
storage landforms in (i) longitudinal (in-stream linkage), (ii) lateral (channel – hillslope relationship), and (iii) vertical (channel
bed – subsurface connection) direction is highly dynamic (e.g., Lane et al., 2017; Fryirs, 2013; Fryirs et al., 2007) and crucial
if sediment from different origins reaches the valley floor and contributes to the glacifluvial transport in the proglacial channel
network of an outwash plain (e.g., Beylich et al., 2009; Brierley et al., 2006). Glacifluvial sediment evacuation is predominant
in the paraglacial period (Church and Ryder, 1972) and is considered as the last transport process of the sediment cascade (e.g.,
Carrivick and Heckmann, 2017; Geilhausen et al., 2012b; Schrott et al., 2003; Chorley and Kennedy, 1971).
Alpine proglacial areas are in general highly dynamic fluvial systems (e.g., Leggat et al., 2015; Micheletti et al., 2015; Baewert
and Morche, 2014; Mao et al., 2014; Gurnell et al., 1999; Warburton, 1990), triggered by daily to seasonal meltwater
fluctuations and high-magnitude/low-frequency events (e.g., Baewert and Morche, 2014; Marren, 2005; Beylich and Gintz,
2004). Combined with the high sediment supply by glacifluvial erosion of glacial diamictic till, braided channels emerge in
direct glacier proximity (e.g., Gurnell et al., 1999; Maizels, 1995). Glacifluvial sediment transport mainly contributes to the
gradual stabilization of proglacial areas (e.g., Carrivick and Heckmann, 2017; Lane et al., 2017; Ballantyne, 2002; Gurnell et
al., 1999). Depending on (i) sediment composition, (ii) runoff variability, (iii) channel slope, and (iv) potential confinement
by, e.g., moraines or (debris-covered) glacier ice, the channel turns into a single thread river with increasing distance to the
glacier terminus (e.g., Gurnell et al., 1999, Maizels, 1995). The transition from proglacial braided to single-thread river
stretches moves upstream against the flow direction parallel to retreating glaciers. This headward erosion process often starts
at a knickpoint in the longitudinal river profile (e.g., Hilgendorf et al., 2020; Schlunegger and Schneider, 2005). Another
dominant process supporting the formation of single channels is riverbed incision when the transport capacity exceeds the
sediment supply (e.g., Wilkie and Clague, 2009; Gurnell et al., 1999). Selective sediment transport (Wilcock and McArdell,
1997; 1993) as a glacifluvial process in formerly glaciated environments creates an infrequently mobile armor layer (Bunte
and Abt, 2001) by grains exceeding the transport capacity, while lateral sediment supply (e.g., channel migration, embankment
failure) remains possible. In headwaters like proglacial reaches, riverbed armoring acts as blankets (type of blockage), which





limits vertical connectivity in different spatial and temporal scales and inhibits the reworking of subsurface sediments. Thus,
an armor layer as a blanket temporarily removes sediment stores from the sediment cascade model. Controlling parameters for
the establishment of blankets are the grain size distribution of the bed material and the transport regime of the channel (Fryirs,
2013), defined by the hydraulic parameter 'flow competence' – the largest particle a flow can move (Benn and Evans, 2013).
Flow competence is mainly impacted by the runoff conditions, which are predicted to change by global warming (e.g., Huss
and Hock, 2018; Farinotti et al., 2012; Braun et al., 2000). The glacier mass of the Austrian Alps is expected to decrease
continuously (Fischer et al., 2018), which implies changes in the future glacial discharge regimes: (i) on a short time scale,
glacial meltwater is predicted to increase due to deglaciation, (ii) in a long-term perspective, the runoff is expected to decrease
by exceeding the expected moment of peak water (Huss et al., 2014; Farinotti et al., 2012). Strongly dependent on the glacier
size, this turning point is predicted before 2050 for European glaciers (Huss and Hock, 2018), upon which the runoff will lose
its glacial characteristic over time with a shift from glacial to nival runoff regimes. Alongside these predictions, reduced runoff
affects the flow competence of proglacial rivers, hence the transport capacity (Pralong et al., 2015), and impacts riverbed
coarsening by glacifluvial erosion.
Channel bed stabilization is one essential step regarding proglacial channel evolution in a diamictic outwash plain. This
glacifluvial process is relevant for adequately describing the downstream sediment yield of proglacial areas. Therefore, this
paper hypothesizes that the glacifluvial erosion process leads to a gradual coarsening of the channel bed material (supported
by non-fluvial, glacial deposited sediment), resulting in the long-term to the vertical disconnection between an erosion-resistant
pavement layer and the diamictic subsurface sediments. This discretization has so far been neglected in the sediment cascade
approach. For this purpose, the proglacial part of the river Möll at the foreland of Austria's biggest glacier Pasterze, was
investigated. Currently, the sediment yield of the Pasterze catchment consists mainly of suspended sediment (Avian et al.,
2018; Geilhausen et al., 2012b), resulting in fine sediment depositions in the downstream located reservoir Margaritze
(Knoblauch et al., 2005; Krainer and Poscher, 1992). Whether this behavior remains the same in the future by changing runoff
characteristics was investigated using predicted runoff by 2050 based on a glacio-hydrological model. A high-resolution digital
elevation model (DEM) was created for hydrodynamic-numerical modeling, and bedload transport formulas were used to
predict the flow competence of the proglacial channel. The ongoing establishment of a pavement layer by grain sizes exceeding
the modeled transport capacity and (exposed) non-fluvial sediment in sections with greater distance to the glacier limits the
connectivity within the glacifluvial system. The results obtained allow a revision and extension of the glacifluvial system of
the sediment cascade approach by incorporating the effects of glacifluvial sediment transport coupled with global warming.



## 2 Study site

### 2.1 Pasterze Glacier

The investigated study reach is in Carinthia in the national park Hohe Tauern at the foreland of the Pasterze (47°5'8" N; 12°43'24" E), the biggest glacier in Austria and the Eastern Alps (16.6 km² in 2012). The 4 km long glacier tongue is characterized by (i) a high mean annual rate of retreat of up to -50 ma$^{-1}$ (Fischer et al., 2018) and (ii) a pronounced debris coverage (approx. 75 % in 2012; Kellerer-Pirklbauer and Kulmer, 2019). The total length loss of the Pasterze Glacier since LIA amounts to -2200 m until 2015 (Fischer et al., 2018). The debris mantle at the glacier tongue's southern part (orographic right) results in a lower ablation rate of up to 35 % by a minimum debris thickness of 15 cm (Kellerer-Pirklbauer et al., 2008). The proximal glacier foreland is characterized by a low gradient (Geilhausen et al., 2012b; Krainer and Poscher, 1992), debris-covered dead ice landforms (e.g., Le Heron et al., 2022; Avian et al., 2018; Geilhausen et al., 2012a; Krainer and Poscher, 1992), diamictic sediment (Geilhausen et al., 2012b) and one main proglacial channel. This glacier-fed river is the major inflow into the reservoir Margaritze, located around 300 m downstream of the catchment outlet (downstream of Sandersee; Fig 1).

### 2.2 Proglacial River

The investigated reach covers around 850 m between the glacier terminus (2100 m a.s.l.) and the inflow (delta area) into the continuously increasing lake 'Pasterzensee' at 2070 m a.s.l. (Fig. 1). This lake evolved around 2010 by a braided river system from 2004 onwards (Avian et al., 2020) upstream of the lake 'Sandersee' (formed in the late 1950s; Krainer and Poscher, 1992). The investigated channel is composed of four distinct sections: (i) the flat headwater near the glacier terminus ($L=$ 200 m; $S_m=$ 0.13 %), (ii) a transition section ($L=$ 70 m; $S_m=$ 2.4 %) into (iii) the canyon (L= 502 m; $S_m=$ 6.5 %), and (iv) the flat outlet (delta; $L=$ 50 m; $S_m=$ 2.6 %) into the lake 'Pasterzensee'. The channel evolved in the ablation season of 2015 at today's beginning of the canyon, verified by images from the automatic camera (Fig. 2) installed at the 'Kaiser-Franz-Josefs-Höhe' (Fig. 1). Almost the entire investigated proglacial channel (except the delta area) is confined by the debris-covered glacier tongue and debris-covered dead ice both characterized by slower melting rates (Kellerer-Pirklbauer et al., 2008). The runoff behavior shows typical glacial characteristics with high summer (up to $Q_{max}=$ 25 m³s$^{-1}$) and low winter runoff (down to $Q_{min}=$ 0.1 m³s$^{-1}$) and strong seasonal and diurnal fluctuations (Geilhausen et al., 2012b; Krainer and Poscher, 1992).



**Figure 1:** Location of the study site: **(a.)** Carinthia, Austria; **(b.)** proximal foreland of the Pasterze Glacier, where the dashed rectangle indicates the proglacial river Möll, including *(1)* glacier tongue (clean), *(2)* glacier tongue (debris-covered), *(3)* Pasterzensee, *(4)* Sandersee, *(5)* reservoir Margaritze, (6) Kaiser-Franz-Josefs-Höhe; **(c.)** study reach, based on the UAV survey, supplemented by the measuring sites for sediment analysis.

**Figure 2:** Investigated proglacial channel: **(a.)** comparison of the flow paths after the channel formation in 2015 and with the UAV survey in 2018; **(b.)** images illustrating the channel formation, recorded by the automatic camera installed at the 'Kaiser-Franz-Josefs-Höhe' (pictures provided by Großglockner Hochalpenstraße).

## 2.3 Sediment budget

Within the exposed proglacial zone of the Pasterze Glacier, more than 23 % of the area accounts for glacifluvial deposits (Geilhausen et al., 2012a), with sediment storages in ice-proximal locations and along the drainage system (Geilhausen et al., 2012b). The glacier foreland with glacifluvial deposited sediment is characterized by moderately well-rounded, poorly sorted (Krainer and Poscher, 1992) glacial diamictic till, including big boulders, gravel, and sand (Fig. 3), which partly covers dead ice landforms (e.g., Le Heron et al., 2022; Avian et al., 2018; Geilhausen et al., 2012a). Near the glacier terminus, kettles and dead-ice holes are frequent landforms, and slopes of ice-cored terraces show great rates of retreat (Avian et al., 2018). The subsystem proximal glacier foreland is decoupled from (i) the active hillslopes subsystem and (ii) the lateral glacier foreland subsystem. While the proglacial area is a dynamic system with a high potential for glacifluvial sediment reworking processes (Avian et al., 2018), the proglacial lakes (Sandersee, Pasterzensee; Fig. 1) act as long-term sediment storages (buffers; Fryirs, 2013) significantly reducing the landform connectivity between glacifluvial erosion and the downstream sediment yield





(Geilhausen et al., 2013). Most of the sediment flux at the outlet of the catchment area is assumed as suspended load
(Geilhausen et al., 2012b) with implications on (i) deposition rates in the continuously expanding proglacial lakes (Avian et
al., 2018; Geilhausen et al., 2013) and (ii) the downstream sediment management of the reservoir Margaritze (Knoblauch et
al., 2005; Krainer and Poscher, 1992).

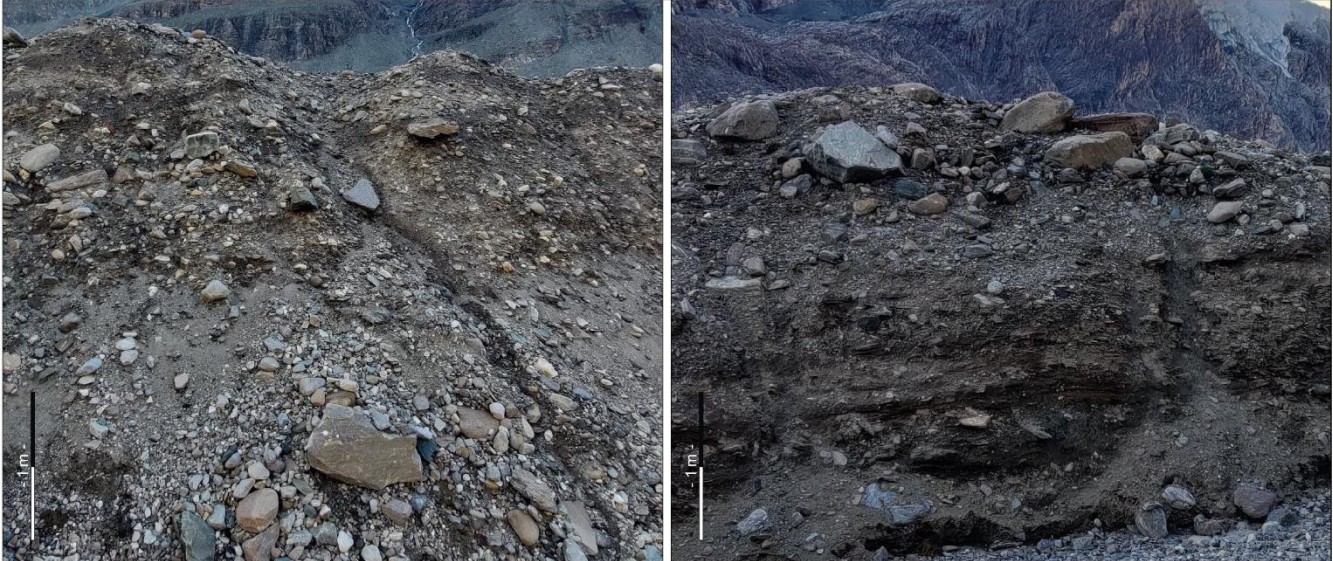

**Figure 3:** River embankment of the investigated proglacial canyon already incised in the poorly sorted diamictic sediment (photographs
taken during the fieldwork).

## 3 Methods

### 3.1 UAV survey

The mapping was carried out during low flow conditions in autumn 2018, where the 850 m long river reach (Fig. 1) was
covered by an unmanned aerial vehicle (UAV; type: hexacopter KR 615) equipped with a compact camera (type: Sony ILCE-
6000; focus length 16 mm) mounted on a stabilized gimbal. The survey was performed in two stages: (i) the entire study area
was covered with a constant flight level of 55 m above the riverbed, and (ii) the canyon in a second flight with a constant flight
level of 20 m above the channel bed (approx. surrounding terrain level). In total, 1371 photos (6000x4000 px) were taken,
whereby a requested overlap of 80 % (forward) and 60 % (sideward) was achieved. Before the flights, ground control points
(GCPs) were placed along the banklines to improve the geodetic accuracy of the digital elevation model (DEM). Due to limited
accessibility and high and steep channel embankments, no GCPs were laid out in the channel. All GCPs were mapped by a
GNSS-RTK device (type: Leica GS18 T).





## 3.2 DEM preparation

In post-processing, the software PhotoScan by Agisoft (version 1.2.6) was used to create (i) a 3D point cloud and (ii) an orthomosaic according to the principle of Structure-from-Motion (Westoby et al., 2012). This approach uses images taken from multiple perspectives to compute a 3D surface based on image-matching algorithms combined with multi-view stereo techniques. This process allows the calculation of the camera position and orientation (Snavely et al., 2008). The mapped GCPs were used for geo-referencing the model and accuracy assessment of the transformation (Fonstad et al., 2013). First, 1371 photos were used in the alignment, the camera position and the orientation of the individual images were estimated, and a sparse point cloud was calculated. The mapped coordinates of ten GCPs were assigned for geo-referencing in the next step. The sparse point cloud was purged to (i) remove high outliers and misaligned points, (ii) optimize the camera position, and (iii) minimize the error between the GCPs. This refinement, including the accuracy assessment by the remaining four GCPs, led to a root-mean-square error (RMSE) of 0.056 m ($X_{RMSE}$= 0.025 m; $Y_{RMSE}$= 0.044 m; $Z_{RMSE}$= 0.024 m). In the third step, the DEM was calculated (3940 points m$^{-2}$) with a ground resolution of 15.9 mm px$^{-1}$, and an orthomosaic (7.9 mm px$^{-1}$) was arranged.

## 3.3 Sediment sampling

The sediment sampling in all accessible sections was done by the line sampling approach (LS) according to Fehr (1987) as the state-of-the-art method for gravel-to-cobble-bed mountain rivers (Lang et al., 2021). Therefore, all grains (b-axis) along the line projection are measured (at least 150 grains). At four characteristic points, line samplings were carried out and mapped with the GNSS-RTK device (circles in Fig. 1). For the inaccessible canyon, the sediment analysis was done by the one-to-one counterpart of the field method (Lang et al., 2021) – the digital line sampling in post-processing on the images taken during the UAV mapping. At all six temporarily non-wetted sediment bars (during low flow conditions) in the canyon (triangles in Fig. 1), the grains were measured manually according to the field method by Fehr (1987). For this purpose, at least 150 grains touched by a virtual line in flow direction were measured (b-axis). The conversion from a relative frequency into a relative volume distribution followed the approach of Fehr (1987). Both applied methods only consider the coarse fractions (partial grain size distribution; Fehr, 1987), which was sufficiently accurate for the objectives of this study. While Fehr (1987) suggests the cut-off at $b \geq 1$ cm, the truncation for adequate identification of grains in the digital line sampling is strongly dependent on the image resolution ranging between $b > 10$ px (Detert et al., 2018) and $b > 20$ px (Purinton and Bookhagen, 2019). The finer fractions are predicted by a Fuller distribution for getting the final grain size distribution of each measurement (Fehr, 1987).

## 3.4 Hydrodynamic-numerical model

A one-dimensional hydrodynamic-numerical model was set up (software Hec-Ras by the United States Army Corps of Engineers) for calculating the hydraulic parameters (i) bed shear stress and (ii) energy gradient, both relevant for the used bedload transport formulas. Therefore, cross-sections (CS) at a 10 m maximum distance were generated from the high-



resolution DEM. The point density was reduced (down to 490 points per CS) by applying the automatic point filter algorithm
with minimum area change. The hydrodynamic-numerical modeling was performed with (i) steady runoff conditions and (ii)
the predicted maximum mean monthly runoff until 2050 ($Q_{m.max.i}$) by Schöner et al. (2013), determined by a 'Glacier Evolution
Runoff Model (GERM)' (Huss et al., 2008). This glacio-hydrological model of the Pasterze Glacier is based on (i) the A1B
scenario according to IPCC (2007), (ii) precipitation in daily resolution, (iii) air temperature in daily resolution, (iv) DEM, and
(v) the glacier edge of 2003 and 2012. Daily climate data from the Sonnblick (3105 m a.s.l.) were applied for model runs in
the past, and bias-corrected data from regional climate model output were used for the scenario run (Loibl et al., 2011). The
model was calibrated using measured glacier mass balances from 2005 to 2012. One of the model output data is runoff in daily
resolution (Schöner et al., 2013).
**3.5 Initiation of motion**
The calculation was done by the formula for the initiation of motion of bedload for steep mountain channels according to
Eq. (1) by Rickenmann (1990), modified by Eq. (2) according to Chiari and Rickenmann (2007) to consider the increased flow
resistance due to macro-roughness elements (e.g., boulders or step-pool sequences) in the canyon ($S_m$= 6.5 %; $S_{max}$= 18.9 %).
$$q_c = 0.065 * \left(\frac{\rho_s}{\rho_w} - 1\right)^{1.67} * g^{0.5} * I_R^{-1.12} * d_{50}^{1.5} \tag{1}$$

Here, the specific discharge ($q_c$), calculated by the maximum velocity ($v_{max.i}$) and maximum depth ($d_{max.i}$) in the CS, is a function
of the characteristic grain diameter ($d_{50}$), the energy gradient ($I_R$), and the ratio between sediment ($\rho_s$) and fluid density ($\rho_w$).
The calculation results of this traditional approach (valid in the flat headwater, transition section, and delta) are termed $d_{50.c:i}$.
$$I_{red} = I_R * \left[\frac{0.133*Q^{0.19}}{g^{0.096}*I_R^{0.19}*d_{90}^{0.47}}\right]^a \tag{2}$$

The reduced energy gradient ($I_{red}$) is calculated by the discharge ($Q$), the energy gradient ($I_R$), the characteristic grain size $d_{90}$
(for which 90 % of the bed material is finer), and $a$= 1.5. The calculation results with $I_{red}$ are labeled with $d_{50.r.i}$ in this study,
valid for the steep canyon with macro-roughness elements. The mean characteristic grain size $d_{90,}$ valid for the entire canyon
and required for this calculation step, was derived from the adjusted Wolman count method (Wolman, 1954), as Hauer and
Pulg (2018) described. According to this field-based method, the b-axis length of the three largest grains in each CS (touched
by the virtual cross-section line) of the canyon was manually measured on the high-resolution aerial images. In total, 171
grains were measured ($b$= 546-3715 mm), corresponding to a mean $d_{90}$= 1290 mm for the entire canyon between CS 552 m
and CS 50 m (Fig. 1).





## 4 Results

### 4.1 Sediment analysis

The sediment analysis shows a downstream coarsening $(d_{50.m:LS.1}= 30$ mm $< d_{50.m:LS.2}= 48$ mm $< d_{50.m:LS.3}= 79$ mm; Fig. 4) with almost the same grain size distribution in the delta area $(d_{50.m:LS.4}= 39$ mm; Fig. 4) as in the headwater. The evaluation of the digital line sampling (six specific points in the canyon) illustrates a much coarser composition $(d_{50.m:DLS.1}= 209$ mm; $d_{50.m:DLS.2}= 233$ mm; $d_{50.m:DLS.3}= 303$ mm; $d_{50.m:DLS.4}= 240$ mm; $d_{50.m:DLS.5}= 223$ mm; $d_{50.m:DLS.6}= 179$ mm; Fig. 4). Large particles were measured in every characteristic point (ranging between $d_{90.m:DLS.6}= 667$ mm and $d_{90.m:DLS.4}= 1225$), and the largest grain size was detected in the steepest part of the entire proglacial channel (CS512; $b= 3700$ mm).

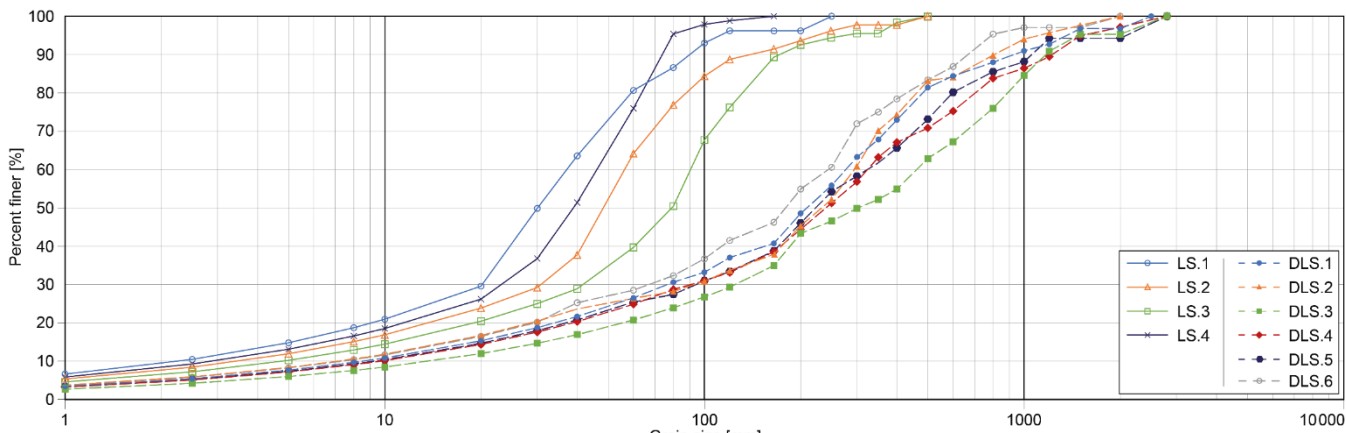

**Figure 4:** Grain distribution curves (corrected according to Fehr, 1987): **(a.)** line sampling (LS; continuous lines) for the headwater and the delta and **(b.)** digital line sampling (DLS; dashed lines) for six specific points in the inaccessible canyon.

### 4.2 Development of flow competence

According to the glacio-hydrological model GERM of the Pasterze Glacier, the maximum mean monthly runoff $(Q_{m.max.i})$ continuously increases in the ablation seasons until June 2035 $(Q_{m.max.2035}= 17.9$ m³s$^{-1}$), following a decrease until 2050 $(Q_{m.max.2050}= 12.7$ m³s$^{-1})$, which is expected to be again around the level of 2018 $(Q_{m.max.2018}= 12.2$ m³s$^{-1}$). In the same period, the maximum mean monthly meltwater runoff $(Q_{m.melt.max.i})$ is predicted to increase until 2034 before decreasing by more than factor two until 2050 $(Q_{m.melt.max.2018}= 4.9$ m³s$^{-1}$; $Q_{m.melt.max.2034}= 9.6$ m³s$^{-1} >> Q_{m.melt.max.2050}= 3.5$ m³s$^{-1}$). The calculated flow competence (characteristic grain sizes $d_{50.c:i}$; $d_{50.r:i}$) runs parallel to the predicted hydrograph, a detailed consideration according to $Q_{m.max.2035,}$ and the grain size measurements $(d_{50.m:i})$ in the longitudinal profile shows two contrary results between (i) the headwater and (ii) the canyon. The maximum calculated characteristic grain size near the glacier terminus (CS 842 m – CS 622 m; $S_m= 0.13$ %; no macro-roughness elements; Fig. 1) by the traditional approach according to Rickenmann (1990) is almost the same as determined on-site by the line sampling (up to $d_{50.c:LS.2}= 49$ mm; $d_{50.m:LS.2}= 48$ mm; Fig. 5a). In the transition section (CS 622 m – CS 552 m) with a slightly increased channel gradient $(S_m= 2.4$ %), a much bigger characteristic grain size was calculated then measured $(d_{50.c:LS.3}= 170$ mm $> d_{50.m:LS.3}= 79$ mm; Fig. 5a). For the flow competence in the steep canyon





(CS 552 m – CS 50 m; $S_m$= 6.5 %; macro-roughness elements), the calculated characteristic grain size according to Chiari and
Rickenmann (2007) with the reduced energy gradient $(I_{red})$ is smaller than measured by the digital line sampling approach
$(d_{50.r:DLS.1}$= 113 mm < $d_{50.m:DLS.1}$= 209 mm; $d_{50.r:DLS.2}$= 131 mm < $d_{50.m:DLS.2}$= 233 mm; $d_{50.r:DLS.3}$= 112 mm < $d_{50.m:DLS.3}$= 303 mm;
$d_{50.r:DLS.4}$= 37 mm < $d_{50.m:DLS.4}$= 240 mm; $d_{50.r:DLS.5}$= 105 mm < $d_{50.m:DLS.5}$= 223 mm; $d_{50.r:DLS.6}$= 72 mm < $d_{50.m:DLS.6}$= 179 mm;
Fig. 5b). The beginning of the canyon was defined around 40 m upstream of the steepest part of the entire proglacial channel
(around CS 512 m; $S_{max}$= 18.9 %), where a knickpoint – a pronounced convexity in the longitudinal channel profile –
developed, characterized by the largest calculated $d_{50.r}$= 326 mm. The calculation results indicate for all specific points in the
canyon that the measured characteristic grain sizes $(d_{50.m:i})$ exceed the calculated flow competence $(d_{50.r:i})$ by the order of 1.8-
6.4 at the maximum predicted discharge in June 2035 $(Q_{m.max.2035})$.

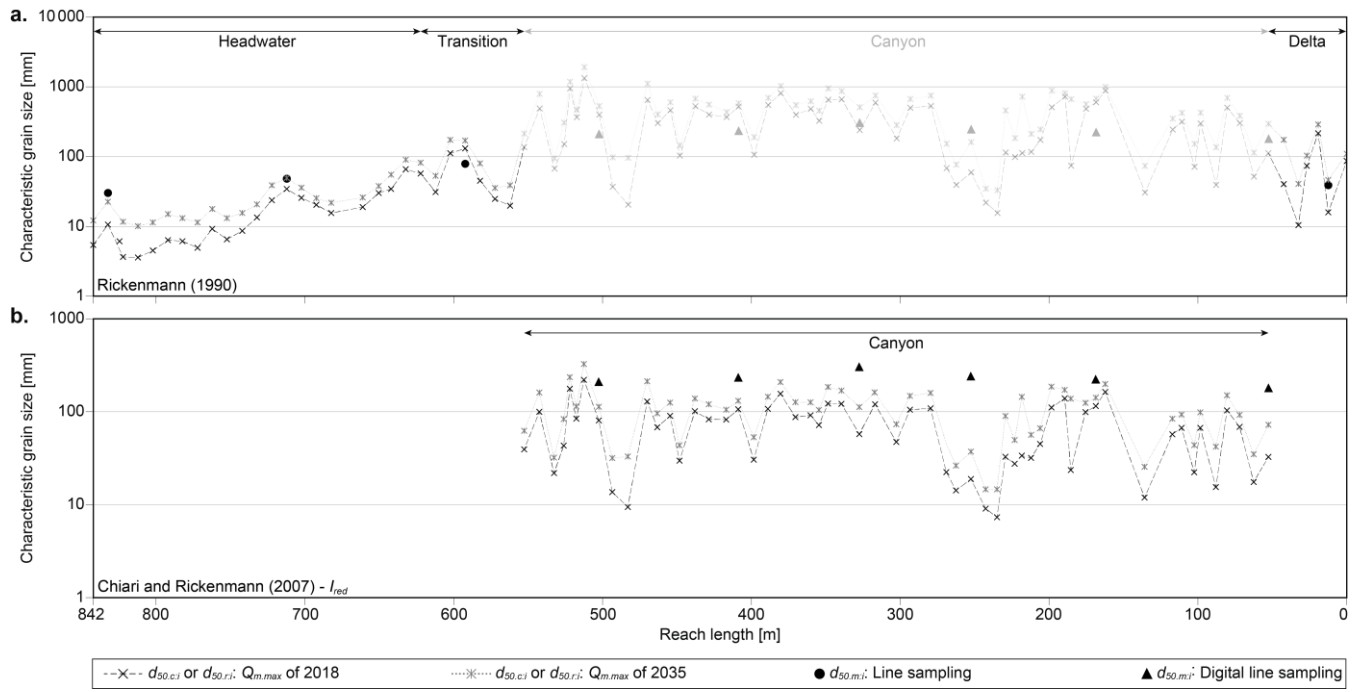


**Figure 5:** Longitudinal course of the calculated characteristic grain sizes (flow competence): **(a.)** approach according to Rickenmann (1990),
where transparently displayed parts of the graph are outside the scope and invalid for the canyon. **(b.)** approach with the reduced energy
gradient $(I_{red})$ – valid for the canyon with macro-roughness elements – by Chiari and Rickenmann (2007). Each graph is supplemented by
the measured characteristic grain sizes $(d_{50.m:i})$ on-site (circle) and those evaluated by the digital line sampling approach (triangle). The results
refer to the predicted maximum mean monthly runoff by 2050 in July 2035 $(Q_{m.max.2035})$ compared to 2018 $(Q_{m.max.2018})$.
**5 Discussion**
**5.1 Channel evolution process**
Glacifluvial sediment reworking is strongly coupled to runoff characteristics (e.g., Leggat et al., 2015; Micheletti et al., 2015;
Pralong et al., 2015; Baewert and Morche, 2014; Mao et al., 2014). Reduced peak meltwater runoff is expected by exceeding





the moment of peak water (e.g., Huss et al., 2014; Farinotti et al., 2012). For European glaciers, this tipping point is forecasted
before 2050 (e.g., Huss and Hock, 2018), which is reflected in the GERM of the Pasterze Glacier (Schöner et al., 2013). The
maximum mean monthly meltwater runoff is predicted for 2034; its decline in the years afterward might be the reason for
almost the same total runoff in 2050 as in 2018 ($Q_{m.max.2018}$= 12.2 m³s⁻¹; $Q_{m.max.2050}$= 12.7 m³s⁻¹). The shift and alteration of the
runoff cause limitations in the bedload transport (Pralong et al., 2015), and greater channel stabilization tendencies by
glacifluvial sediment reworking are given with increasing distance to the glacier terminus (e.g., Carrivick and Heckmann,
2017; Lane et al., 2017; Ballantyne, 2002; Gurnell et al., 1999). Channel bed incision by glacifluvial erosion is a stabilization
process (e.g., Wilkie and Clague, 2009; Gurnell et al., 1999), leading to gradual coarsening of the channel bed substrate. This
development is supported by the change in the channel gradient (Gurnell et al., 1999). Separated by a knickpoint (e.g.,
Hilgendorf et al., 2020; Schlunegger and Schneider, 2005), the flat headwater ($S_m$=0.13 %) in direct glacier proximity
transitioning to the incised canyon ($S_m$=6.5 %). This knickpoint is defined by the highest gradient ($S_{max}$=18.9 %; CS 512) of
the entire investigated proglacial reach and is located around the source of the proglacial river in 2015 (Fig. 2). The analysis
of the sediment composition and the hydrodynamic-numerical model results tend to the potential for riverbed incision in the
headwater and for pavement layer formation (channel bed stabilization) in the canyon.
The dominant process in the headwater is headward erosion, already known from, e.g., a fluvial drainage basin in Switzerland
(Schlunegger and Schneider, 2005). Starting from the point with the highest gradient (knickpoint; $S_{max}$), the continuous
glacifluvial erosion shifts the knickpoint more upstream (Hilgendorf et al., 2020). First indicators of this development were
detected up to 140 m upstream of the knickpoint (CS 512 m) in the transition section (CS 650 m – CS 550 m; Fig. 1), defined
(i) by a much bigger flow competence (largest particle a flow can move) than in the headwater and (ii) the exposure of already
very big grain sizes ($b$> 2000 mm). Like in the canyon, fine fractions are expected to be transported continuously out of the
headwater, which gives the tendency of a progressive channel bed armoring by sediment coarsening (Bunte and Abt, 2001;
Dietrich et al., 1989). Exactly this glacifluvial development is already occurring in the steeper canyon. In the past, the local
sorting of the diamictic sediment by glacifluvial erosion resulted in channel bed incision (Fig. 6), a dominant process during
and after deglaciation (Gurnell et al., 1999). The calculation results, according to the approach with $I_{red}$ (Chiari and
Rickenmann, 2007), valid for torrential flow characteristics with macro-roughness elements (e.g., Pralong et al., 2015; Nitsche
et al., 2011; Chiari and Rickenmann, 2011), indicate pavement layer formation in the canyon. Previous studies have already
observed less bedload transport at the foreland of the Pasterze Glacier (Avian et al., 2018; Geilhausen et al., 2012b).
The progressive armoring by (i) glacifluvial erosion combined with (ii) decreasing flow competence in the long-term
perspective (e.g., Huss and Hock, 2018; Pralong et al., 2015; Huss et al., 2014; Farinotti et al., 2012) will establish an erosion-
resistant pavement layer, composed of glacifluvial deposits of the armor layer, and additionally supported by (exposed) non-
fluvial sediments (Hauer and Pulg, 2020; Bunte and Abt, 2001). In contrast to the infrequently mobile armoring layer (Fryirs,
2013; Bunte and Abt, 2001), this development will inhibit channel bed incision. This trend was already observed in specific
points in the canyon (triangles in Fig. 1), where very coarse sediment composition (up to $d_{90.m:DLS.4}$= 1225 mm; Fig. 4) and
occasionally non-fluvial sediment were measurable. These locations indicate the assumption of limited channel bed incision
in the future (Fig. 6). For rivers characterized by such glacial deposited non-fluvial sediment, Hauer and Pulg (2018)
implemented the term glacial-till cascade, which contributes remarkably to channel bed stabilization.

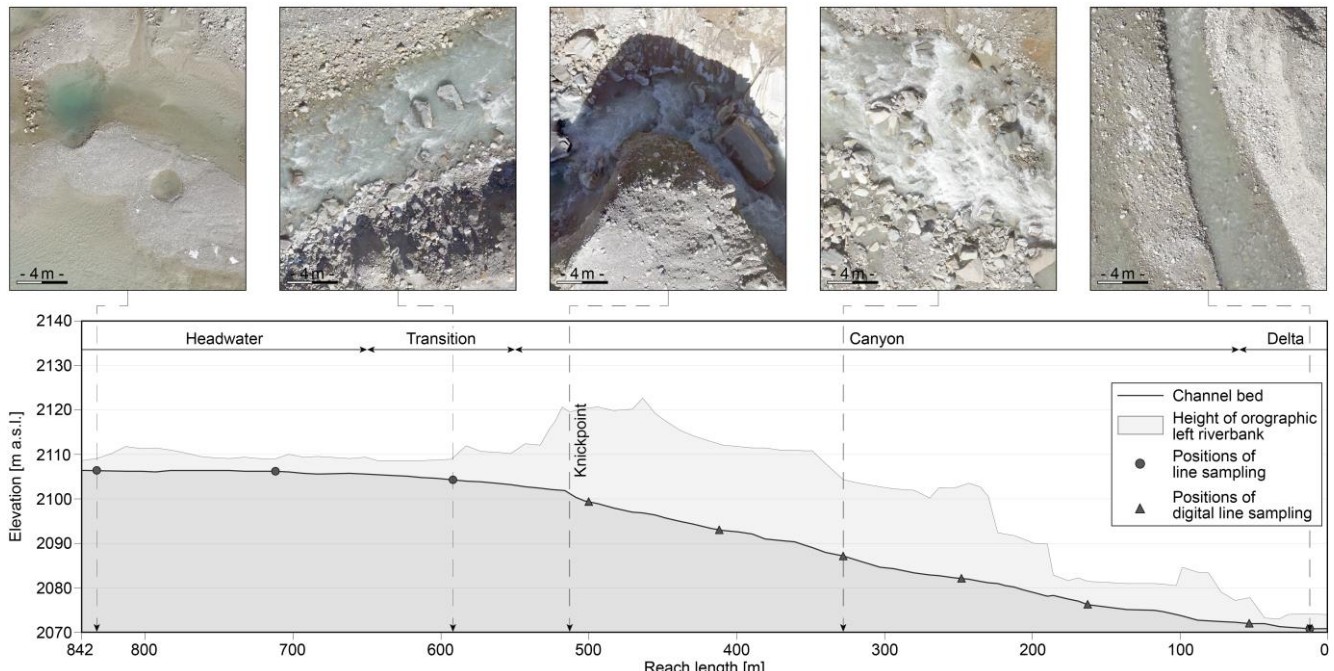

**Figure 6:** Longitudinal section of the investigated reach length highlighting (i) the different river sections and (ii) the specific points for the
analysis in the canyon (compare Fig. 1). The pictures taken by the UAV show the different sediment composition in some characteristic
points (same scale).
Sediment transport out of proglacial areas by glacifluvial erosion is predominant (Church and Ryder, 1972) and generalized
as the last transport process of the proglacial sediment cascade model, controlled by the erosion location and flow velocity
(regulators) if sediment is deposited or glacifluvial evacuated (e.g., Carrivick and Heckmann, 2017; Geilhausen et al., 2012b).
However, the study results show a more differentiated picture resulting in a more precise and refined proglacial sediment
cascade model by the glacifluvial system (dotted and grey highlighted frames in Fig. 7). The gradual evolution of proglacial
rivers starts as braided channel network in direct glacier proximity and transits to single-thread rivers with greater distance to
the glacier terminus (e.g., Gurnell et al., 1999, Maizels, 1995). This novel longitudinal differentiation in the sediment cascade
approach enables the implementation of two new in-stream storage types: (i) grain sizes of the already (partly) established
infrequently mobile armor layer (e.g., Fryirs, 2013; Bunte and Abt, 2001) that exceed the transport capacity form together with
(ii) exposed non-fluvial deposits an erosion resistant pavement layer. This extension and refinement require the implementation
of the new regulator 'grain size composition' in the subsystem 'glacier', decisive if grains are transported glacifluvial
downstream or support the gradual proglacial channel bed stabilization as a non-fluvial deposit (Fig. 7). In contrast to an
infrequently mobile armoring layer (i) limiting vertical connectivity (e.g., Fryirs, 2013; Fryirs et al., 2007; Brierley et al., 2006)
and (ii) removes temporally sediment storages from the sediment cascade (Fryirs, 2013), an established pavement layer acts
as an erosion-resistant blanket, disconnecting the linkage between the (stabilized) proglacial channel bed and the





unconsolidated diamictic sediment in the subsurface. Pavement layer formation by glacifluvial erosion is thus an essential
stabilization process and is part of the well-known landform decoupling (e.g., Fryirs, 2013; Fryirs et al., 2007; Brierley et al.,
2006). As the sediment cascade model shows decoupled subsystems in the Pasterze catchment (Geilhausen et al., 2012b), the
pavement layer is composed of the new in-stream storage types: (i) big glacifluvial deposits (grain sizes exceeding the transport
capacity forming an armor layer) and (ii) non-fluvial glacial deposits. If all subsystems of a sediment cascade model are
coupled with each other, supraglacial debris (Geilhausen et al., 2012b) and coarse colluvial deposits can also be contained in
both new in-stream sediment storage types and contribute to channel bed stabilization. However, rivers in proglacial areas with
an established pavement layer still enable lateral sediment supply, often triggered by high-magnitude/low-frequency events
(e.g., Baewert and Morche, 2014; Marren, 2005; Beylich and Gintz, 2004).

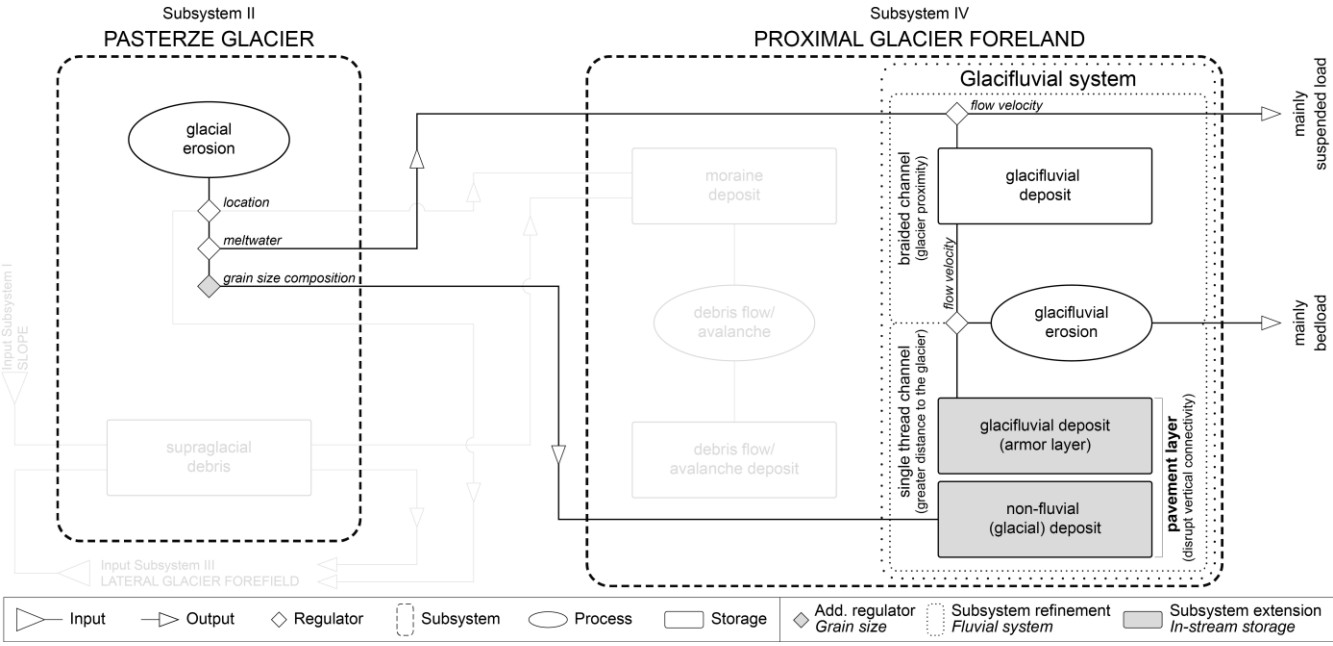

**Figure 7:** Refinement (dotted frames) and extension (grey highlighted in-stream sediment storage types) by the glacifluvial system within
subsystem IV (proximal glacier foreland) of the conceptual model of the sediment cascade approach for the Pasterze landsystem. The
evolving pavement layer (blanket) by sediments in these additional in-stream storage types will disrupt the vertical connectivity between the
stabilized proglacial channel bed and the subsurface diamictic deposits (Fryirs, 2013). Due to the decoupled subsystems in the catchment
(Geilhausen et al., 2012b), the transparently outlined connections only complete the sediment cascade model but are irrelevant to the
objectives of this study. Modified after Geilhausen et al. (2012b).
Measurements in high mountain areas are prone to uncertainties (Avian et al., 2020), as (i) inaccessibility and (ii) torrential
flow characteristics lead to limitations in the (iii) geometry and calibration data acquisition as well as in sediment sampling.
Due to inaccessibility (canyon) and low flow conditions during the measurements, the water surface corresponds to the channel
bed in the hydrodynamic-numerical model, and representative sediment analysis of the canyon (digital line sampling) could
be done in the temporary non-wetted area (possible due to the strongly pronounced diurnal discharge cycle). However, the
steep channel gradient and the very low discharge during the UAV mapping allow this kind of application with minimal
uncertainties in the hydraulic predictability of sediment movement. Moreover, it can be assumed that the same grain size





composition is present in the permanently wetted main channel, although it will probably be already coarser due to constant
exposition to glacifluvial erosion. In both applied approaches, line and digital line sampling, small grain sizes are often
underestimated (Fehr, 1987; Purinton and Bookhagen, 2018). However, a thorough analysis reveals that a reduced
characteristic grain size $d_{50.m:i}$ by up to 40 % (greater consideration of the underestimated smaller grain sizes) is still lower than
the calculated flow competence according to the maximum mean monthly runoff by 2050 ($Q_{m.max.2035}$= 17.9 m³s⁻¹) with the
reduced energy gradient ($I_{red}$) according to Chiari and Rickenmann (2007). This detailed evaluation further emphasizes the
tendency towards pavement layer formation. The importance of considering energy losses by macro-roughness elements to
achieve more plausible results was pointed out in various studies (see literature in Chiari and Rickenmann, 2011) for bedload
transport calculations, especially in steep mountain streams ($S$> 4-6 %; Badoux and Rickenmann, 2008). Comparative analyses
with bedload transporting events in high-alpine torrents indicate an overestimation by up to factor 10 with traditional bedload
formulas (e.g., Nitsche et al., 2011; Chiari and Rickenmann, 2011). Regarding the Glacier Evolution Runoff Model (GERM),
uncertainties are given by the climate evolution and input data quality (Huss et al., 2014; Schöner et al., 2013).

## 5.2 Drivers for future proglacial channel avulsion

Glacifluvial sediment reworking of glacial deposits reduces landform connectivity, which is relevant for sediment storage or
export (e.g., Fryirs, 2013; Fryirs et al., 2007; Brierley et al., 2006). Landform and subsystem connectivity is highly dynamic
(Lane et al., 2017), and changes are often triggered by high-magnitude/low-frequency events (e.g., Baewert and Morche, 2014;
Marren, 2005; Beylich and Gintz, 2004). Furthermore, the connectivity in the catchment and reach scale can be in- and
decreased by randomly distributed dead ice landforms and ice-cemented sediments, coexisting parallel to un- and metastable
sediment deposits in proglacial areas (Gärtner-Roer and Bast, 2019). In contrast to flood-driven river avulsion (e.g., Slingerland
and Smith, 2004; Jones and Schumm, 1999; Brizga and Finlayson, 1990), proglacial channel avulsion may be caused by the
downwasting of dead ice landforms (e.g., Benn and Evans, 2013; Lukas et al., 2005). However, in a long-term perspective
with decreasing runoff (e.g., Huss and Hock, 2018; Farinotti et al., 2012), laterally migrating channels do not impact a fully
developed erosion-resistant pavement layer of an older river stretch. Furthermore, ongoing glacifluvial erosion in new channels
leads to riverbed coarsening, resulting again in the gradual establishment of an erosion-resistant pavement layer.
The presence of dead ice at the forefield of the Pasterze Glacier has long been known (Lagally, 1932). Consequently, different
dead ice landforms like hummocky moraines, ice-cored moraines, or kettles could be detected in the past (e.g., Le Heron et
al., 2022; Avian et al., 2018; Seier et al., 2017; Geilhausen et al., 2012a; Krainer and Poscher, 1992). The continuous increase
in debris cover at the Pasterze Glacier (Fischer et al., 2018) is one prerequisite for dead ice formation (Gärtner-Roer and Bast,
2019) following decoupling of the active upglacier (Benn and Evans, 2013) and slower melting rates (Keller-Pirklbauer et al.,
2008). Melting of debris-covered ice in proximal locations of the Pasterze Glacier leads to rapidly changing (landform) surface
conditions (Geilhausen et al., 2012a). The constant observation of the study area confirms this statement and assumes the
formation of a new channel in the center of the valley with well-known glacifluvial processes following up. Investigating the
gradual proglacial channel (network) evolution in response to melting dead ice landforms is highly relevant (i) in general for





describing future proglacial channel development and quantifying proglacial sediment yields and (ii) in particular for the high-
alpine reservoir management downstream of the investigated reach.

**6 Summary and Conclusion**

This paper predicts the future flow competence (the largest particle a flow can move) of the proglacial part of the river Möll
according to the glacio-hydrological model GERM (glacier runoff evolution model) of the Pasterze Glacier by 2050. Due to
the diamictic characteristic of the outwash plain and the predicted runoff variability triggered by global warming, the study
results indicate a needed distinction between an infrequently mobile channel bed armoring and an erosion-resistant pavement
layer. This is an important definition and an essential post-glacial development process, which has been widely neglected up
to now in defining proglacial channel evolution stages. The following outcomes need to be highlighted:
(1)    While recently deglaciated river sections are prone to glacifluvial headward erosion (against flow direction parallel to the

glacier retreat) due to the fine sediment composition near the glacier terminus, river sections with a greater distance to

the glacier are characterized by sediment coarsening of the bed material. This gradual process will inhibit further channel

bed incision by establishing an erosion-resistant pavement layer composed of (i) grain sizes exceeding the transport

capacity and (ii) exposed non-fluvial deposits. Triggered by global warming, the short-term increase and long-term

decrease of the flow competence will reinforce this stabilization process, contrasting with an infrequently mobile

armoring layer. This development is considered as an important (up to final) stage in proglacial river evolution.

(2)    The calculation results indicate the tendency of a pavement layer formation in the canyon, which allows an extension and

refinement of the glacifluvial part within the subsystem 'valley floor' of the proglacial sediment cascade model: (i)

braided channels in direct glacier proximity differ from (ii) sections with a stabilized and erosion-resistant channel bed

(in the long-term) in increasing distance to the glacier terminus. This pavement layer is defined by the two new in-stream

storage types (armor layer composed of glacifluvial deposits and non-fluvial sediment). This development leads to

vertical landform decoupling between the erosion-resistant channel bed and the diamictic sediment in the subsurface.

(3)    In the long-term perspective, river avulsion driven by the melt-out of (buried) dead ice bodies will mainly contribute to

the stabilization in the catchment and reach scale. Investigating the channel evolution in response to melting dead ice

bodies is highly relevant for quantifying future sediment dynamics of proglacial areas transitioning from glacial to non-

glacial landscapes.

**Data availability**

All the experimental data used in this study are available from the authors upon request.



## Author contribution

MP and CH planned and designed the research. MP, PF, AN, GW, and BH performed the investigation, data curation, and evaluation. MP and CH did the original draft preparation and visualization with equal contributions from all co-authors. All authors were part of the review and editing of the manuscript.

## Competing interests

The authors declare that they have no conflict of interest.

## Disclaimer

Publisher's note: Copernicus Publications remains neutral with regard to jurisdictional claims in published maps and institutional affiliations.

## Acknowledgment

This paper was written as a contribution to the Christian Doppler Laboratory for Sediment Research and Management. In this context, the financial support by the Christian Doppler Research Association, the Austrian Federal Ministry for Digital and Economic Affairs and the National Foundation for Research, Technology and Development is gratefully acknowledged. Moreover, the authors thank Rolf Rindler and Martin Fuhrmann for supportive fieldwork and Johann Aigner for discussions on sediment transport dynamics.

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
