# Peer review of "Channel evolution processes in a diamictic glacier foreland. Implications on downstream sediment supply: case study Pasterze / Austria"

_Hydrology and Earth System Sciences, 2023_

## Author Comment (AC2)

**Reply on anonymous Reviewer #1**

I am afraid that this paper is either fundamentally flawed, or there are critical steps in the explanation of what is done that are not explained. This needs to be addressed before a full review is possible.

The work in the paper is based upon the Rickenmann (1990) proposal of a formula for estimating the critical (specific) discharge required for sediment transport (qc). This was modified in Rickenmann (2006) and vulgarized more widely in Chiari and Rickenmann (2007, Table 1, Equation 7, modified as the authors note) to treat the effects of a reduced energy slope given macroroughness elements in streams. The critical (specific) discharge is commonly combined with an actual discharge (q) in a threshold-based sediment transport equation (Chiari and Rickenmann (2007), Table 1, Equation 6) to estimate time-varying sediment transport capacity (i.e. the transport rate is a f (q-qc)). The authors have taken some future predictions of discharge (q) and want to see how the sediment transport competence changes as the glacier shrinks and goes through "peak water". This is all a very legitimate thing to do, well justified by the literature which the authors know and use well. The idea that a river erodes and sorts its bed as a glacier retreats and so progressively stabilizes is a good working hypothesis for known river response following glacier retreat; although the authors don't quite pick up on wider knowledge regarding the temporality of this process as argued in the work of Marren and others (i.e. a river erodes close to the glacier but then deposits eroded sediment further downstream causing an erosion-aggradation response).

> *Reply: Thank you for the appreciative words on the idea of the paper, the working hypothesis, the literature review/knowledge, and the discussion of the results. Thank you for critically reviewing the submitted manuscript (MS).*
>
> *We agree that the initial version of the manuscript needs a revision regarding the points the reviewer made. A detailed discussion of the comments can be found in the following section. We believe that we have considered all comments appropriately.*

**1. General comments**

**A)** But, this is where I get lost and this is not helped by Section 3.5 which is poorly explained. First, when they present their results, they compare the "characteristic grain sizes" with those measured (e.g. Figure 5). Of course, if you know qc and its changes through time, you can invert the Rickenmann type equations to get a characteristic grain size. But you have to know qc and you don't know it or how it will evolve. My only explanation of what they have done here is that they have completely misunderstood the associated equations. In the paper, at L199, the authors define qc as the specific discharge whereas qc is actually the **critical** specific discharge, that which the q must exceed for transport to occur. I fear that they have taken their future discharge scenarios

(some measures of q) as qc and then estimated the associated grain-size. That is, in their Eq. (1) they have used q and not qc. This is completely flawed. Now I may have mistaken something here, but to make it clear, if q is being used instead of qc, this is a very basic mistake that makes the analysis meaningless. If I am mistaken it is because how they go from q to qc to a characteristic grain-size is not explained in the paper sufficiently.

*Reply: (1) As the reviewer correctly pointed out, the condition $q>q_c$ is required for the initiation of motion of bedload. We used Eq. (1) according to Rickenmann (1990) in the inverse form to calculate the characteristic grain size $d_{50}$, which is in any case permissible, which was also considered by the reviewer.*

*(2) Our goal in the paper was to calculate the largest characteristic grain size $d_{50}$ with the highest predicted discharge by the glacio-hydrological model GERM. In other words, we have calculated the $d_{50}$ at the maximum predicted specific discharge $q_{max}$.*

*(3) To make it easier understandable, we added new indices to the relevant terms of Eq. (1), which is shown in the original version below.*

$$q_c = 0.065 * \left(\frac{\rho_s}{\rho_w} - 1\right)^{1.67} * g^{0.5} * I_R^{-1.12} * \boldsymbol{d_{50}^{1.5}} \tag{1}$$

*In our paper, we use the assumption: $q_c = q_{max}$, whereby $q_{max}$ is calculated by Eq. (2).*

$$q_{max.i} = v_{max.i} * t_{v.max.i} \tag{2}$$

*Here, the maximum specific discharge per cross-section ($q_{max.i}$) is calculated by the maximum flow velocity ($v_{max.i}$) and the corresponding depth ($t_{v.max.i}$) in each cross-section. This calculation step was already mentioned in the initial MS (LN 200).*

*When we are using now the inverse Eq. (1), we calculated in a way the critical characteristic grain size $d_{50.crit.i}$ for the given maximum specific discharge $q_{max.i}$. Thus, the Eq. (3) looks like:*

$$d_{50.crit.i} = \left[\frac{1}{0.065} * \frac{q_{max.i}}{\left(\frac{\rho_s}{\rho_w}-1\right)^{1.67} * g^{0.5} * I_R^{-1.12}}\right]^{\frac{2}{3}} \tag{3}$$

*This means, the calculated characteristic grain size $d_{50.crit.i}$ is theoretically the critical grain size for the initiation of motion for the given predicted maximum specific discharge $q_{max.i}$ according to the glacio-hydrological model GERM.*

*To illustrate the entire calculation approach here in the reply, Eq. (4) by Chiari and Rickenmann (2007) is also presented. This modification is needed to consider the increased flow resistance due to macro roughness elements in the canyon (LN 197 and 203-206 in the initial MS).*

$$I_{red} = I_R * \left[\frac{0.133 * Q^{0.19}}{g^{0.096} * I_R^{0.19} * d_{90}^{0.47}}\right]^a \tag{4}$$

*The reduced energy gradient ($I_{red}$) is used in Eq. (3) instead of $I_R$ for the CS in the canyon.*

*(4) To illustrate that the approach described above has been used correctly, Eq. (1) was applied in its original form for this revision. After the characteristic grain size $d_{50.m.i}$ (indices 'm' for 'measured') is available from the line sampling (headwater and delta) and digital line sampling*

*(canyon), the critical discharge $q_c$ can be calculated for the given measured $d_{50.m.i}$. These results show, that $q_{c.i} > q_{max.i}$ for all grain size distributions and for all discharges used (see Tab. 1). This analysis was also done with predicted maximum mean daily discharge (highest resolution possible by GERM) until 2050 (modeling period). Since $q_c$ is – among others (see Eq. 1) a function of $I_R$ (energy gradient), $q_c$ for $d_{50.m}$ is changing between the used discharges.*

**Tab. 1:** *Comparison of the results between the critical specific discharge ($q_{c.i}$) for the measured $d_{50.m}$ by Eq. (1) – modified by Eq. (4) – and the calculated maximum specific discharge ($q_{max.i}$) according to Eq. (2) for the observation years 2018, 2035 (highest predicted mean monthly discharge) and 2045 (predicted maximum mean daily discharge).*

| **River station** [m] | $q_{c.2018}$ [m³/sm] | $q_{max.2018}$ [m³/sm] | $q_{c.2035}$ [m³/sm] | $q_{max.2035}$ [m³/sm] | $q_{c.2045}$ [m³/sm] | $q_{max.2045}$ [m³/sm] |
|---|---|---|---|---|---|---|
| *502* | 7.532 | 1.804 | 6.145 | 2.457 | 5.556 | 3.596 |
| *409* | 9.302 | 2.873 | 8.795 | 3.744 | 33.878 | 4.054 |
| *327* | 25.265 | 2.094 | 12.196 | 2.762 | 11.865 | 3.792 |
| *252* | 51.943 | 1.155 | 9.968 | 1.872 | 5.267 | 2.418 |
| *168* | 6.921 | 2.574 | 6.238 | 3.176 | 5.455 | 4.109 |
| *52* | 23.674 | 1.842 | 7.956 | 2.337 | 3.121 | 1.292 |

*This table clearly shows – despite the given study limitations (LN 326-344 in the initial MS) – the maximum specific discharge by the maximum mean daily discharge according to GERM ($q_{max.2045}$) is still smaller than $q_c$ for the corresponding $d_{50.m}$. ($q_{c.2045}$).*

**B)** Second, Eq. (1) shows that the reduced energy slope (Ir) and the grain-size (D50) drive qc. Erosion/deposition will lead to the sediment sorting that drives changes in Ir and D50 and hence qc. This is not addressed in the modelling as far as I can see. To address this you would need a time-dependent sediment sorting treatment that also took into account changes in subglacial sediment export. Sediment sorting is at its most intense when capacity is greater than supply (subglacial sediment export, sediment supply from banks) and so how the critical discharge and the competence will evolve has to take into account supply as well. The authors recognize this in the introduction and the discussion but none of their analysis actually simulates this process, as far as I can see.

*Reply: (1) We agree with the reviewer, that a changing geometry (by erosion/deposition) changes $q_c$. We will add to the study limitations (LN 326-344) in the revised MS, that the prediction model is a steady-state 1D hydraulic model.*

*(2) Sediment supply is a relevant parameter for the process (e.g., Bathurst, 2007; Rickenmann et al., 2006), but is not considered in the calculation approach for the initiation of motion of bedload. It determines the specific discharge for a given characteristic grain size – or it is used inversely (as it was done in the presented study) to calculate a critical characteristic grain size for a defined specific discharge.*

**C)** Third, I should add that there is likely one other fundamental flaw. The analysis is done for the maximum mean monthly runoff and the hydrological model is daily; but the actual maximum daily discharge will be substantially greater than this due to diurnal discharge variation. Any analysis of this kind would need to work with downscaled daily discharge data as and used the mean maximum monthly runoff calculated from an hourly timescale. This diurnal discharge variation is also likely to change significantly as the glacier declines in size.

> *Reply: (1) The input data of the glacio-hydrological model GERM (LN 189 in initial MS) are in daily resolution resulting in output data is mean daily discharge (Huss et al., 2008).*
>
> *(2) The input data in the hydraulic model (HEC-RAS) is the maximum mean monthly runoff, calculated by the output of GERM in daily resolution.*
>
> *(3) For this review, the calculations were done for the maximum predicted mean daily discharge until 2050 (end of the modeling period) – see Tab.1 for the results.*
>
> *(4) Correct, the diurnal discharge variation is predicted to change as the glacier size declines. Less amount of the meltwater discharge in the total discharge is predicted after the exceedance of the moment of peak water. According to the data by GERM, this turning point is predicted in the 2030s according to the applied A1B climate scenario (LN 227 in the initial MS).*

**D)** The discussion is well-situated in the literature but deviates substantially from the results that are provided. Indeed, very few results are provided in the paper.

> *Reply: Thank you for this comment. We have a different opinion, as the applied methods provide substantial results for discussion (which is well situated in the literature, annotated also by the reviewer) and the conclusions are valid.*

The authors should attend to the following more minor issues if they can resolve the above flaws, redo the analysis, and then resubmit the paper.

**2. Specific comments**

**L29** "by" should be "with"

> *Reply: Thank you, we changed the wording as suggested.*

**L35** "exceeds the geological norm"; not clear

> *Reply: Thank you for this comment. The sediment yield declines towards the geological norm during the paraglacial period (Ballantyne, 2002). We change the wording during the revision.*

**L47** "last" in what sense?

> *Reply: Glacifluvial sediment transport is the last process of a sediment cascade of a spatially defined area (e.g., catchment)*

**L50** what is "triggered"?

*Reply: This is not an appropriate term in this context. Thus, we rephrase this sentence during the revision.*

**L56** "to" should be "from"

*Reply: Thank you, we changed the wording as suggested.*

**L58** "parallel to …" should be "as glaciers retreat"

*Reply: Thank you, we changed the wording as suggested.*

**L64** "blankets" is a poor term

*Reply: This is a well-known term describing landform (dis)connectivity: "Bed armor acts as a blanket that inhibits the reworking of subsurface sediments." (Fryirs et al., 2007; Church et al., 1988).*

**L73** "by …" poorly phrased

*Reply: In literature, different terms/phrases are used to describe this turning point, like 'expected moment of peak water' by Schaefli (2015) or just 'peak water' by Huss and Hock, 2018.*

**L74** but it depends on the glacier

*Reply: Correct, the expected moment of peak water depends on the glacier size, but this turning point "has already been reached or passed or is expected to occur in the coming two or three decades" (Huss and Hock, 2018 and literature therein).*

**L84-85** but have you measured bedload sediment?

*Reply: (1) Yes, we measured bedload sediment. Depending on the discharge, flow velocity, and the expression of the armor layer, the threshold grain size between suspended sediment and bedload is 1 mm in alpine rivers (Maniak, 2005).*

*(2) With the applied sediment sampling approaches, only bedload sediment can be measured. The applied state-of-the-art field method for gravel-to-cobble-bed mountain rivers is line sampling (Fehr, 1987; described in LN169-181 in the initial MS). This approach only considers coarse fractions with a cut-off at b≥ 1 cm. In the digital line sampling, the image resolution is decisive for truncation – which ranges between b> 10 px (Detert et al., 2018) and b> 20 px (Purinton and Bookhagen, 2019).*

*(2) That mainly suspended sediment is deposited in the downstream located reservoir Margaritze is known from literature (Knoblauch et al., 2005; Krainer and Poscher, 1992; LN 85 in initial MS).*

**L107-109** does not make sense as written

>*Reply: This is a statement by Avian et al., 2020: "From 2004 onwards, a braided river system evolved into a second lake (Pasterzensee), which was established in 2010."*

**L131** "kettle-holes"

>*Reply: Correct, kettle-holes can be found at the proglacial area of the Pasterze Glacier. We corrected it as suggested.*

**L138** "on" should be "for"

>*Reply: Thank you, we corrected it as suggested.*

**L172-173** be clearer here that you measured the grain-size off digital images where the canyon was inaccessible. You also need to explain how you guaranteed the equivalence of grain-sizes from the line-sampling in the field which measured b-axes and the line-sampling of the imagery which measures surficial exposure of grain-sizes. See also **L208-209** – the b axis measured on an image is not the same as the true b axis – there is a bias – and one that increases as a function of the level of sediment reworking.

>*Reply: (1) Thank you for this comment. The difference in the grain size composition between the headwater and canyon is clearly visible in Fig. 6 in the initial MS. Finding the true b-axis can be also challenging in the field, especially in proglacial areas with randomly distributed very big, often non-fluvial grain sizes (see Hauer and Pulg, 2018). Thus, finding the true b-axis is not only an uncertainty in the digital line sampling or automatic image analyzing software.*
>
>*To guarantee the equivalence between line sampling in the field and digital line sampling on the high-resolution images, reference samples from the canyon are missing due to inaccessibility (LN 328 in initial MS and other replies). But if the measured $d_{50.m}$ by the digital line sampling is reduced by 20 % (in parts by up to 30 %), $q_{max.2045}$ is still too low for the initiation of motion.*
>
>*(2) Sediment reworking is limited according to the calculation results.*

**L182 and onwards** – the topography you use here will not be the river bed – how did you deal with this in your cross-sections? The same issue also applies to the digital grain-size survey; how do you get the grain-sizes for underwater zones?

>*Reply: (1) The UAV photogrammetry was carried out during low flow conditions in the morning at the end of the ablation season (LN 146 in the initial MS). Due to the inaccessibility (canyon) for cross-sectional terrestrial survey, the water surface during the UAV photogrammetry corresponds to the channel bed in the hydraulic model. However, the steep channel gradient and the low discharge during UAV photogrammetry allow this kind of application with minimal uncertainties in the hydraulic predictability of bedload movement (stated in the study limitations – LN 331 of the initial MS).*

*(2) According to the approach by Fehr (1987), the virtual line for the digital line sampling was drawn in the flow direction near the permanently wetted area. Thus, the digital line sampling was done in the temporarily non-wetted area. So, it can be assumed that the same grain size composition is present in the permanently wetted main channel, although it will probably be already coarser due to constant exposition to glacifluvial erosion (stated in the study limitations – LN 333 in the initial MS).*

---

## Author Comment (AC3)

**Reply on anonymous Reviewer #1**

Overall, the manuscript: *"Channel evolution processes in a diamictic glacier foreland. Implications on downstream sediment supply: case study Pasterze / Austria"* is well prepared and addresses relevant scientific questions on the future channel evolution processes in such systems. The submission is well structured, and the language is fluent and precise. However, when reviewing the manuscript some questions arose (general comments), which need to be addressed/considered during the review of the manuscript.

> *Reply: Many thanks to the Reviewer for the appreciative words regarding scientific importance, structure, and language. Thank you for critically reviewing the submitted manuscript (MS).*
>
> *We agree that the initial version of the manuscript needs a revision regarding the points the reviewer made. A detailed discussion of the comments can be found in the following section. We believe that we have considered all comments appropriately.*

My main concern is the assumption of a static system to predict future morphological processes in a fully dynamic system. I agree with the authors that different processes can be seen and explained, but interpretations and conclusions are in my opinion associated with uncertainties.

> *Reply: Thank you for this valuable comment. We agree with the reviewer, that proglacial areas are dynamic systems (stated in LN 49 in the initial MS), and uncertainties are given (stated in the study limitations; LN326-344 in the initial MS). However, the statements are indeed valid, because:*
>
> *(1) Glacial diamictic till, characterized by unsorted to poorly sorted sediment with grain sizes ranging in size from clay to boulders (Harland et al., 1966; LN 40 in the initial MS), is present in the proglacial area of the Pasterze Glacier (LN 129 in the initial MS). This common glacially deposited material (Benn & Evans, 2013) is prone to selective glacifluvial sediment transport, where grains exceeding the transport capacity form an infrequently mobile armor layer (Bunte & Abt, 2001; LN 63 in the initial MS). Glacially deposited non-fluvial boulders will have a big contribution to channel bed stabilization (e.g., Hauer and Pulg, 2018) and vertical landform decoupling (Fryirs et al., 2013), a conclusion of the presented study (LN 307; Fig. 7 in initial MS).*
>
> *(2) Although the channel is formed in a highly dynamic environment (LN 49 in the initial MS), the lateral confinement by dead ice and ice-cemented sediment led to the deep canyon section due to the slower ablation rate of debris-covered ice (by up to 35 %; Kellerer-Priklbauer et al., 2008; LN 100 in the initial MS). Melting debris-covered (dead) ice will lead to a broadening of the canyon in the future, resulting in greater wetted width, which implies less transport capacity and bed shear stress for bedload mobilization. Furthermore, a stabilized channel bed by grains exceeding the transport capacity and glacially deposited non-fluvial boulders will remain erosion resistant.*
>
> *(3) Moreover, the calculation was done with the predicted maximum mean monthly discharge within the modeling period of the applied glacio-hydrological model GERM (for this revision, the*

*calculation was also done for the maximum mean daily discharge until 2050 – see later reply). The results of GERM follow the well-known prediction of the expected moment of peak water (e.g., Huss and Hock, 2018; Schaefli, 2015). This turning point, where the meltwater discharge will decrease after the exceedance of this turning point due to less glacier ice volume, is predicted for European glaciers before 2050 or has already passed (LN65 in the initial MS; Huss & Hock, 2018). The results by GERM (according to the A1B scenario) predict the maximum meltwater discharge for the 2030s ($Q_{m.melt.max.2034}= 9.6\ m^3s^{-1}$) with a decreasing mean monthly meltwater runoff until the end of the modeling period in 2050 (LN 228 in the initial MS). This statement is also valid for the maximum mean daily runoff until 2050 (best resolution possible by GERM; see later reply).*

*(4) That the observations, results, and key findings of this study are transferable to other proglacial areas is described in a reply below.*

**1. General comments**

**A)** The authors obtained a DEM in 2018 and used predicted runoffs for steady-state 1D simulations (2018, 2035, and 2050), which means the system (topography, roughness) was static over time. However, from the figures, it becomes evident, that the system is fully dynamic and the canyon part changed drastically within the last three years. Hence, I am wondering how reliable these predictions are, as there are no morphological changes considered.

> **Reply:** *Thank you for this valuable comment. We are aware that we have modeled the flow competence with a steady-state 1D simulation. In terms of a prediction model, uncertainties are given (LN326-344 in initial MS), but reliable results are provided for discussion.*

**B)** What was the reason that the authors have chosen runoff in 2035, whereas the maximum mean monthly meltwater runoff is predicted for 2034?

> **Reply:** *(1) Output of glacio-hydrological model GERM is the discharge in different ‚reservoirs'* (term used in GERM) – in the case of Pasterze Glacier the reservoirs (i) 'ice', (ii) 'snow', (iii) 'surface runoff', (iv) 'groundwater' and (v) 'permafrost', each in (mean) daily resolution (Schöner et al., 2013). This differentiation allows the statement, that the peak meltwater runoff of glacier ice is predicted for the 2030s according to the A1B scenario ($Q_{m.melt.max.2034}= 9.6\ m^3s^{-1} > Q_{m.melt.max.2035}= 4.9\ m^3s^{-1}$), although the summation of all reservoirs leads to maximum predicted mean monthly discharge for 2035 ($Q_{m.max.2035}= 17.9\ m^3s^{-1} > Q_{m.max.2034}= 15.2\ m^{3-1}$).*

> *(2) The statement that peak ice melt is predicted for the 2030s according to the A1B scenario is also visible in the data of maximum mean daily meltwater discharge (highest resolution possible by GERM) of glacier ice until 2050: $Q_{d.melt.max.2034}= 16.9\ m^3s^{-1} > Q_{d.melt.max.3035}= 8.7\ m^3s^{-1}$ and all other mean daily meltwater discharge by glacier ice until 2050.*

**C)** The authors write about the torrential flow characteristics, which may lead to morphological changes, but use mean monthly values for their steady-state simulations. Hence, I am wondering, if such mean values can replicate the morphological system, or if important runoff peaks, leading to a morphological development of the system, are missing. The authors write: "Landform and subsystem connectivity is highly dynamic (Lane et al., 2017), and changes are often triggered by high-magnitude/low-frequency events." Especially short-time events may lead to a break-up of the bed armoring layer.

> **Reply:** *(1) Thank you for this comment. We meant with 'torrential' that the river type (especially of the canyon) is 'torrent-like'. We corrected this term.*
>
> *(2) During the revision, we did the modeling with the predicted maximum mean daily runoff (best resolution possible by GERM) until 2050 (predicted for July 2045; $Q_{d.max.2045}$= 28.02 m³s⁻¹; according to the A1B scenario). As it is visible in the table below, it is obvious, that the calculated $d_{50.cirt.r}$ is bigger than calculated by the maximum mean monthly runoff (2035), but still smaller than the measured $d_{50.m}$ by the digital line sampling. Thus, the conclusion of the presented paper – the development of a pavement layer by grains exceeding the transport capacity and glacially deposited non-fluvial boulders – is still valid with the maximum predicted discharge in the best resolution possible by GERM.*

**Tab. 1:** *Comparison of the $d_{50.m}$ (by digital line sampling – DLS) and calculated $d_{50.r}$ (with reduced energy gradient $I_{red}$ according to Chiari & Rickenmann, 2007) for the respective years.*

| River station [m] | $D_{50.m}$ [mm] | $D_{50.crit.r.2018}$ [mm]* | $D_{50.crit.r.2035}$ [mm]* | $D_{50.crit.r.2045}$ [mm]* |
|---|---|---|---|---|
| 502 | 209.128 | 80.482 | 113.280 | 156.207 |
| 409 | 232.729 | 106.117 | 131.443 | 152.754 |
| 327 | 302.941 | 57.461 | 112.329 | 172.094 |
| 252 | 239.863 | 18.914 | 37.235 | 122.245 |
| 168 | 222.786 | 114.986 | 141.777 | 184.109 |
| 52 | 179.396 | 32.615 | 71.823 | 139.820 |

*\*The indices has changed in response to a comment of reviewer 2.*

> *(3) We agree with the reviewer, that an armoring layer can be broken up by e.g., high-magnitude/low-frequency events. Although such extreme events are not predictable by models, this rarely possible situation is mentioned in the initial MS (LN 51 or 317).*

**D)** The authors write about bed armoring and that within the canyon an armor layer has already developed. Have the authors considered the status of the armor layer and if it was fully developed? A fully developed bed armor layer can be calculated depending on the grain size distribution, e.g. given by Günter (1971).

> **Reply:** *(1) Thank you for this comment. The started development of an armoring layer is an assumption from the authors, as the measured $d_{50.i}$ (in 2018) is already bigger than calculated by the maximum mean monthly discharge in 2018 ($Q_{m.max.2018}$= 12.2 m³s⁻¹; see table before). The specific critical discharge $q_c$ (calculated by Eq.1, modified by Eq.2; see initial MS) necessary*

*for the movement of the corresponding $d_{50.m}$, is higher than the actual maximum specific discharge in 2018.*

*(2) Calculation approaches for the development of an armor layer (e.g., Little and Mayer, 1976; Knoroz, 1971, Günter, 1971 or Jäggi, 1983), sediment data from the surface and subsurface is necessary. As the canyon was inaccessible during the fieldwork, no sediment data of the subsurface is available.*

*(3) However, the glacial diamictic till does not have a clearly defined/differentiated subsurface sediment (as we know it from armored lowland rivers; e.g. Hunziker and Jaeggi, 2002; Bunte and Abt, 2001), because very coarse grains and even non-fluvial boulders occur randomly distributed over the entire thickness of the glacial diamictic till (Harland et al., 1966).*

**E)** Even though the authors show many references, I am wondering how transferrable the results are to other glacier regions in Europe or even worldwide. The authors write that there are many boundaries involved, such as slope, bed material compositions, but a statement on that should be given.

*Reply: Thank you for this comment.*

*Yes, the results are transferrable to other proglacial areas because:*

*(1) Slope: The channel gradient is an essential controlling parameter for river channel pattern and is one essential hydraulic parameter for calculating the initiation of motion of bedload (Church, 2002). Thus, slope is a relevant hydraulic parameter but not a boundary for a specific study case.*

*(2) Sediment: Diamictic sediment (unsorted to poorly sorted sediment with grain sizes ranging in size from clay to boulders; Harland et al., 1966) is very common in glacially influenced depositional environments (Benn and Evans, 2013). That means, the sediment composition like in this study area can be found in many different other proglacial areas of the Alpine (e.g., Le Heron et al., 2021; Lane et al., 2017; Morche et al., 2012; Carrivick and Rushmer, 2009) and Arctic recently deglaciated areas (e.g., Tomczyk et al., 2020; Rachlewicz, 2009).*

*(3) Confinement: The channel (especially the canyon section) is confined by (buried) dead ice (LN 102, LN 114, LN 130 in the initial MS), which exists in many proglacial areas in addition to ice-cemented sediment (Gärtner-Roer and Bast, 2019). Debris-covered glacier surface, a prerequisite for dead ice development (Gärtner-Roer and Bast, 2019; Benn & Evans, 2013), exists at the Pasterze Glacier (Kellerer-Pirklbauer et al., 2019; Fischer et al., 2018) – mentioned in LN 98 and Figure 1b in the initial MS – and is increasing at many European glaciers (Lardeux et al., 2016) and glaciers worldwide (Mayr & Haag, 2019). Thus, dead ice or ice-cemented sediment can be confinement landforms in many proglacial areas.*

**F)** It was not clear to me if newly generated sediments, as a result of glacier melt, are considered in future predictions. These fine sediments may alter the morphology. Here a statement given by the authors: "Combined with the high sediment supply by glacifluvial erosion of glacial diamictic till,.." Here another question arose, how is the suspended sediment transport and the interaction with the bed considered in the study?

> *Reply: Bedload is responsible for channel bed stabilization and the development of an armor layer. Therefore, suspended sediment (for alpine rivers – grain diameter <1 mm; Maniak, 2005) was not considered in the presented research design.*
>
> *(2) The calculation approaches for the initiation of motion of bedload do not consider suspended sediment.*

**G)** The chapter on the hydrodynamic model needs more details, such as information on the chosen roughness or if a calibration was performed.

> *Reply: Thank you for this comment, additional information will be added to the MS:*
>
> *(1) Based on the measured sediment composition, sensitivity analysis, and literature (e.g., Naudascher, 1992), three different roughness coefficients $k_{st}$ were used: (i) headwater and delta: $k_{st} = 28 \ m^{1/3}s^{-1}$; transition and canyon: $k_{st} = 18 \ m^{1/3}s^{-1}$.*
>
> *(2) Due to the inaccessibility of the canyon, the measurement of calibration parameters (e.g., discharge, water levels) was not possible. Thus, we performed a sensitivity analysis comparing different hydraulic parameters (e.g., flow velocity, shear stress, Froude number, and wetted width) to define the adequate roughness coefficient for the hydraulic model.*

**H)** I recommend having a separate chapter on the Glacier Evolution Runoff Model (GERM), and more details on the calibration of the model (bias-corrected data, regional climate model) and the output. Here my question is why daily values ("One of the model output data is runoff in daily resolution") are used and not e.g. values with an hourly resolution, or maybe three hours maximum.

> *Reply: We will add additional information about GERM to the MS:*
>
> *(1) Precipitation in the catchment has been extracted from a gridded precipitation dataset (Hofstätter et al., 2013) using the mean value of 4 neighboring grid points and by applying a vertical precipitation gradient. For the scenario runs, data from regional climate model runs of the reclip:century simulations (Loibl et al., 2011) have been bias-corrected to the station dataset. For temperature, a monthly bias correction and for precipitation, the quantile-mapping has been applied to adjust the distribution of the climate model output to the measured one.*
>
> *(2) The best resolution of the glacio-hydrological model GERM is discharge in daily resolution. This is a widely used tool for predicting runoff evolution (e.g., Huss et al., 2014; Sorg et al., 2014; Farinotti et al., 2012; Huss et al., Huss et al., 2008).*

**I)** Page 10, line 214: "The sediment analysis shows a downstream coarsening". I think this is misleading, as in the delta the same grain sizes are visible as in the headwater. In general, I think we see here typical morphological patterns, where the different grain sizes depend on the boundaries, such as slope. As the canyon has the highest slope it is evident, that the coarsest sediments can be found there. I think here it is necessary to dig into the data of the HEC-RAS model to get more insight into the hydraulics of the system.

> ***Reply:*** *We agree with the reviewer, that 'downstream coarsening' is a misleading term in this case. We changed the wording with the coarse sediment composition of the canyon as a reference for the description of the finer sediment composition in the headwater and delta.*

**J)** Page 10, line 242: "the maximum mean monthly runoff." As mentioned before, a mean value may not be representative for predicting morphology and channel evolution, as the "Glacifluvial sediment reworking is strongly coupled to runoff characteristics."

> ***Reply:*** *(1) As written in a reply before and presented in Tab. 1, it is evident that the calculation results ($d_{50.crit.r.i}$) with the predicted maximum mean runoff in daily resolution (best resolution possible by GERM) are bigger than with maximum mean monthly runoff, which leads to a smaller difference to the measured $d_{50.m.i}$.(see Tab. 1)*
>
> *(2) As stated as well in the reply before, the key message and conclusion of the paper are still valid with the highest predicted discharge in the best resolution possible by GERM.*

**K)** Page 10, lines 233-235: The paragraph needs some modification and may be rewritten. The authors introduce three discharges: $Q_{m.melt.max.2018}$= 4.9 m³s-1; $Q_{m.melt.max.2034}$= 9.6 m³s-1 >> $Q_{m.melt.max.2050}$= 3.5 m³s-1) and use Q2034 for comparisons. I think here only comparisons between the sampled sediments and the Q2018 can be made, as it is unsure what the channel and the grain size distribution will look like in 2034 or 2050. Hence, I think the statement "In the transition section (CS 622 m – CS 552 m) with a slightly increased channel gradient (Sm= 2.4 %), a much bigger characteristic grain size was calculated then measured ($d50.c:LS.3$= 170 mm > $d50.m:LS.3$= 79 mm; Fig. 5a)" is not valid, as no measurements are available yet. Why is 2050 not included in the figure? Can the authors maybe draw lines of a fully developed bed armor layer? Here I think it needs to be statistically proven that the modified approach by Chiara and Rickenmann led to better results.

> ***Reply:*** *(1) As mentioned in a previous reply, the input discharge for the hydraulic model was not only the predicted maximum meltwater discharge ($Q_{m.melt.max.i}$), but it was the maximum predicted total mean monthly discharge by GERM ($Q_{m.max.2035}$).*
>
> *(2) The approach in the presented paper is a prediction model, which is a widely used tool also in river morphology studies. Although uncertainties are given (see study limitations; LN 326-344 in initial MS), the results are reliable, and the conclusion is valid.*

*(3) The calculation results with the predicted discharge from 2050 are not included, as the goal of the study was to find the highest predicted discharge for comparison with sediment data from the actual situation. The predicted maximum mean monthly runoff in 2050 is almost the same as in 2018 ($Q_{m.max.2018}$= 12.2 m³s⁻¹; $Q_{m.max.2050}$= 12.7 m³s⁻¹; mentioned in LN 226 and 258 in the initial MS) resulting in similar calculation results for $d_{50.crit.r.i}$.*

*(4) A statistical test is not applicable, as reference data (e.g., measurement data) are not available due to missing measurement stations. However, as it is stated in the study limitations of the initial MS (LN 226-344) and as it is stated in literature (see later reply), the consideration of macro-roughness elements and increased flow resistance in steep high-alpine rivers is a relevant modification of bedload calculation approaches for more plausible results.*

**L)** Page 12, line 267: "The analysis of the sediment composition and the hydrodynamic-numerical model results tend to the potential for riverbed incision in the headwater and for pavement layer formation (channel bed stabilization) in the canyon." This estimate is in my opinion only valid for a static system.

> ***Reply:*** *As mentioned in a previous reply, we applied a prediction model for estimating the future flow competence of the proglacial river using predicted discharge data of the glacio-hydrological model GERM. Prediction models are associated with assumptions and uncertainties (see study limitations; LN 326-344 in the initial MS). However, under given conditions, the results are substantial for discussion and the conclusion is valid.*

**M)** Page 14, line 317: "However, rivers in proglacial areas with an established pavement layer still enable lateral sediment supply, often triggered by high-magnitude/low-frequency events." This is evidence that the system will change over time.

> ***Reply:*** *(1) Lateral sediment supply often leads to the broadening of the channel resulting in less bed shear stress, especially in combination with the exceedance of the moment of peak water (see previous reply and LN 75 in the initial MS). Furthermore, lateral sediment supply does not change the composition of the already (partly) stabilized channel bed by grain sizes exceeding the transport capacity and glacially deposited non-fluvial boulders (see previous reply).*

> *(2) As mentioned in a previous reply, the calculation was done with the predicted maximum mean monthly discharge (during the revision also with maximum mean daily discharge – highest resolution possible by GERM). Thus, the calculation was done with the maximum discharges predicted and available for this study area. Extreme events like high-magnitude/low-frequency events cannot be predicted.*

**N)** Page 14, line 326- : The authors write about shortcomings and uncertainties, e.g., in geometry and calibration data acquisition as well as in sediment sampling, but do not quantify it.

> ***Reply:*** *Uncertainties in areas without any measured reference data are difficult to quantify. Thus, uncertainties can only be estimated by e.g., a sensitivity analysis. Therefore, we reduced the measured $d_{50.m}$ by 20 % (in parts up to 30 %) and $q_{max.2045}$ is still too low for the initiation of motion.*

**O)** Page 16, line 369: "This paper predicts the future flow competence (the largest particle a flow can move) of the proglacial part of the river Möll according to the glacio-hydrological model GERM (glacier runoff evolution model) of the Pasterze Glacier by 2050." I think this sentence is misleading, as there were many uncertainties not considered. Hence, I would rather say an estimate.

> ***Reply:*** *Measurements in high mountain areas and especially at glacier forelands are challenging (e.g., weather, accessibility, discharge behavior) as written before and mentioned in the initial MS (LN 326-344). Thus, we agree with the reviewer and changed the wording.*

**P)** Page 15, line 340: "The importance of considering energy losses by macro-roughness elements to achieve more plausible results was pointed out in various studies (see literature in Chiari and Rickenmann, 2011) for bedload transport calculations, especially in steep mountain streams (S> 4-6 %; Badoux and Rickenmann, 2008)." But when looking at Figure 5, I have the impression that the approach given by Rickenmann (1990) fits better than the approach with the reduced energy gradient (Fig. 5b) in the canyon. See my previous comment.

> ***Reply:*** *(1) As mentioned in a previous reply, in the study limitations of the initial MS (LN 326-344) and as stated in the relevant literature (e.g., Chiari and Rickenmann, 2011; Nitsche et al., 2011; Badoux and Rickenmann, 2008; Bathurst, 2007; Chiari and Rickenmann, 2007; Rickenmann et al., 2006), increased flow resistance due to macro-roughness elements must be considered in the calculation approach for more plausible and realistic results.*
>
> *(2) Moreover, in steep high-alpine streams, the form resistance (e.g., by step-pool sequences) – included in Eq2.; see initial MS – can amount to up to 80 % of the total flow resistance (Canovaro et al., 2004).*
>
> *(3) Steep high mountain streams are characterized by (i) wide grain size distribution, (ii) large boulders that remain immobile during high discharges, (iii) step-pool morphology, and (iv) low relative flow depths. All these features lead to additional roughness and flow resistance (Chiari and Rickenmann, 2011; Nitsche et al., 2011). All these characteristics are given for the studied canyon section: (i) glacial diamictic till with grains ranging in size from clay up to boulders, (ii) glacially deposited non-fluvial boulders, (iii) step-pool morphology and low relative flow depths related to the big boulders.*

**2. Specific comments**

**Page 1, line 17:** I would replace "sediment was sampled" by "sediment properties were obtained"

> *Reply: Thank you, we accept the suggestion.*

**Page 1, line 18:** I would not talk here about bedload transport formulas, as it was the one from Rickenmann used and an approach considering the energy line.

> *Reply: We agree with the reviewer, and revised it as suggested.*

**Page 1, line 18:** "Due to the fine sediment composition near the glacier terminus (d50< 79 mm)." I would not call these sediments fine sediment composition.

> *Reply: We agree with the reviewer, the definition of 'fine sediment' is 'particles smaller than 2 mm'. Thus, we changed the wording of the entire paragraph and used the coarse grain size composition of the canyon as a reference value in this study (a "finer sediment composition than in the canyon was measured for the headwater and delta").*

**Page 4, line 116:** "and strong seasonal and diurnal fluctuations." Can the authors give an approximate?

> *Reply: (1) As stated in the initial MS (LN 115-116), seasonal discharge fluctuations range between $Q= 0.1\ m^3s^{-1}$ in winter and up to $Q= 25\ m^3s^{-1}$ in summer (Krainer & Poscher, 1992). As no discharge measurements are available, the daily fluctuations can only be estimated from reconstructed data of the inflow into the Margaritze reservoir, which are of the same order of magnitude as the seasonal fluctuations.*

**Page 9, line 190:** The authors only used the A1B scenario according to IPCC in their study. This needs to be justified.

> *Reply: Field measurements in high alpine environments regarding river morphology and hydraulic modeling are prone to uncertainties (e.g., weather conditions, inaccessibility, torrent-like discharge conditions) and lead to some limitations (e.g., geometry and calibration data, sediment sampling; LN326 – 344 in the initial MS). Limitations are also given in the glacio-hydrological model GERM (LN343-344 in the initial MS). The main uncertainties next to the uncertainty of the future climate evolution are in the calibrated parameter set and in the model (Farinotti et al., 2012) of the glacio-hydrological model. Due to missing winter mass balance measurements in the study area, parts of the calibration of the glacio-hydrological model have been carried out using glaciers at close distance but outside the study region. The glaciers considered are in the same mountain range of the Hohe Tauern region. However, to be able to predict glaciofluvial processes, the A1B scenario according to IPCC was used, which assumes balanced land-use changes and balanced progress across all resources and technologies which represents a middle way between different scenarios.*

**Page 9, line 191:** (v) the glacier edge of 2003 and 2012. What was the glacier edge in 2018 when the field survey was conducted?

> ***Reply:*** *Input data for the glacio-hydrological model GERM are, among others, glacier edges with corresponding digital elevation models of the entire catchment. This data combination was only available for 2003 and 2012 and is therefore used in the model GERM.*